# First Case of Infective Endocarditis Caused by *Vibrio metschnikovii*: Clinico-Diagnostic Complexities and a Systematic Literature Review

**DOI:** 10.3390/clinpract15070118

**Published:** 2025-06-25

**Authors:** Alessandro Carrozzo, Vittorio Bolcato, Luigi Martinelli, Ferdinando Dodi, Antonella Vulcano, Giuseppe Basile, Livio P. Tronconi

**Affiliations:** 1Maria Cecilia Hospital, GVM Care & Research, 48033 Lugo, Italy; 2Department of Cardiac Surgery, ICLAS, GVM Care & Research, 16035 Rapallo, Italy; 3Maria Beatrice Hospital, GVM Care & Research, 50121 Firenze, Italy; 4UOC Microbiologia e Banca Biologica, National Institute of Infectious Diseases, L. Spallanzani, IRCCS, 00163 Rome, Italy; 5Department of Biomedical Sciences and Public Health, University “Politecnica delle Marche” of Ancona, 60124 Ancona, Italy; 6Department of Health and Life Sciences, European University of Rome, 00163 Rome, Italy

**Keywords:** infective endocarditis, *Vibrio metschnikovii*, case report, diagnostic challenge, culture-negative infective endocarditis, systematic review

## Abstract

Background: Non-cholera Vibrio species are rare waterborne pathogens that can cause severe infections. Among these, few cases of *Vibrio metschnikovii* infections have been reported, especially in the gastrointestinal tract, with no cardiac tissue involvement as a result. Following the PRISMA checklist, we conducted a literature review, and thirteen articles for twenty-two cases overall were included: seven cases of sepsis (in three cases, the echocardiographic results were negative), seven cases of pneumonia, two skin infections, eleven cases of diarrhoea, and a gastroenteritis outbreak. This report documents the expanding clinical spectrum and the role played by *V. metschnikovii* in infective endocarditis. Case report: A 28-year-old male patient was referred to the cardiac surgery unit for urgent mitral valve replacement due to suspicion of infective endocarditis. Microbiological tests yielded negative results. Following recovery and discharge with antimicrobial therapy for 6 weeks, the patient experienced prosthesis detachment, necessitating re-hospitalisation for an emergency valve replacement. *Vibrio metschnikovii* was identified on the prosthesis valve through PCR and successfully treated with ciprofloxacin. However, a spontaneous rupture of the ascending thoracic aorta led to a neurological injury. Discussion: This case represents the first case of valve infection caused by *Vibrio metschnikovii*, characterised by diagnostic and therapeutic challenges and the involvement of the great vessels. Also considered in this case, for a disease with a median age of 58 years (11–83) and a male-to-female ratio of 2.2, were one male neonate and six cases for whom neither sex nor age was indicated. Excluding gastrointestinal cases, the septic forms are associated with high morbidity, although the single case described involved a young and healthy subject. Risk factors for the pathogen or predisposing/pathological conditions for endocarditis did not emerge. The routes and the time of infection could not be determined, deepening the possibility of occupational exposure via the patient’s position as a boat worker. Poor sensitivity to third-generation cephalosporins has been reported in the literature: the absence of an antibiogram does not allow for a comparison, although resolution was achieved with ciprofloxacin. Conclusion: The rising global incidence of non-cholera Vibrio infections, driven by environmental changes, calls for urgent research into the factors behind their pathogenicity and infection routes. Diagnostic complexities have emerged together with clinical severity.

## 1. Introduction

Non-cholera strains of the genus *Vibrio* are rare waterborne pathogens that can cause severe infections in humans, particularly in immunocompromised individuals [1]. *Vibrio metschnikovii* is a catalase-positive bacillus typically associated with aquatic environments, specifically brackish water, seafood, animals, and some of the indigenous aquatic organisms in these ecological environments [2,3,4,5]. The incidence of *Vibrio metschnikovii* in enclosed marine systems and estuaries is expected to increase over time due to rising oceanic water temperatures resulting from climate change [6]. Such environmental changes may lead to an increased risk of virulent strains in raw oysters and other types of seafood. The rise in global seafood consumption further accentuates the potential for non-cholera *Vibrio* infections, highlighting the need for awareness and diagnostic vigilance among at-risk populations. *Vibrio metschnikovii*, although less commonly discussed compared to *Vibrio vulnificus* and *Vibrio parahaemolyticus*, can still pose significant health risks [7]. Human infections are extremely rare, with few cases reported in the literature [8]. Gastroenteric involvement appears to be predominant, although septic forms have also been reported, often with an unclear route of infection and no evidence of endocardial involvement [8].

Infective endocarditis (IE) remains a rare condition but one with high morbidity and mortality. The epidemiology of IE has changed owing to the ageing population, the increasing use of implantable cardiac devices and heart valves, and the rise in multidrug-resistant organisms [9]. The causative microorganisms of infective endocarditis vary depending on the site of infection, the source of bacteraemia, and other risk factors associated with the site of entry (e.g., illicit endovenous drugs), but, overall, *Streptococci* spp. and *Staphylococcus aureus* cause 80–90% of cases [10]. *Enterococci* spp., Gram-negative bacilli, HACEK microorganisms (*Haemophilus* spp., *Aggregatibacter actinomycetemcomitans*, *Cardiobacterium hominis*, *Eikenella corrodens*, and *Kingella kingae*), and fungi cause most of the remaining cases of endocarditis.

Some microorganisms require longer incubation in blood cultures; others require serodiagnosis (e.g., *Coxiella burnetii*, *Bartonella* spp., *Chlamydia psittaci*, *Brucella* spp.), and there are some (e.g., *Legionella pneumophila* and *Tropheryma whippelii*) that require special culture media or PCR [11,12].

Unconventional microorganisms with growth selectivity and/or difficulty or atypical ones require continuous infectious disease investigation and reporting [13], mainly considering epidemiological trends [14]. Moreover, a significant change in the antimicrobial regimen, outpatient parenteral antibiotic treatment, and indications for surgery must be considered [11,15]. Complications can be life-threatening and include valvular degeneration, heart failure, systemic embolization with stroke or splenic infarction, abscess formation, and sepsis, and their occurrence and severity are strictly related to prompt adequate treatment [16].

The aim of this study was to present a case of mitral valve endocarditis caused by *Vibrio metschnikovii*, as well as to conduct a systematic review to compare this case with examples from the existing literature, explore the role of the pathogen in the cardiovascular district, and determine the possible route of infection and environmental prevalence, highlighting the diagnostic challenges and clinical severity.

## 2. Case Report

We report the case of a 28-year-old male who works as a boat maintenance worker in the Italian region of Liguria. The Case Report (CARE) checklist was followed and is available as a Appendix A. Personal protective equipment, including safety gloves, safety shoes, head covers, and masks, was available and used during work activities by the subject. The patient had a history of smoking, denied illegal drug consumption and medication intake, and had no previous medical history, nor a predisposing condition for infective endocarditis, recent travel abroad, or injuries. The patient underwent routine cardiology evaluations for work fitness, which consistently demonstrated normal valvular function, and there was no family history of mitral valve disease. In January 2024, the patient was admitted to the emergency department of a local hospital with fever and muscle pain. He was diagnosed with pericarditis and treated with anti-inflammatory medications (prednisone 25 mg per day with a gradual dose tapering), leading to a resolution of his symptoms after a few days.

However, a few weeks later, the patient developed recurrent high fever, chills, and general discomfort, prompting another admission to the local emergency room. Routine blood tests showed a significant increase in inflammatory markers. A transoesophageal echocardiography (TEE) was performed, revealing the presence of multiple vegetations on the anterior mitral leaflet, projecting into the left atrium (Figure 1). An empirical but reasoned antimicrobial therapy was started with daptomycin (700 mg intravenously per day) and ceftriaxone (1 g intravenously per day) after collecting the blood culture samples, according to the decision of an infectious disease specialist.

However, the patient rapidly deteriorated to acute heart failure and was referred to our cardiac surgery centre for emergency treatment. Mitral valve removal with mechanical prothesis implantation, after median sternotomy approach, was performed. The intraoperative aspect of the mitral valve showed multiple lesions which encompassed the valve along both its atrial and ventricular sides. After native valve removal, additional local irrigation with rifampicin solution was performed. The blood cultures, the cultures of the removed native valve, and sternum wound swabs came back negative for common pathogens. However, it should be noted that antimicrobial therapy was started while the previous blood cultures taken in the emergency room were also negative. After 10 days of hospitalisation in the ICU, the patient’s clinical condition was improved, and he was finally discharged. An antibiotic therapy regimen with daptomycin and ceftriaxone intravenously for six weeks was prescribed at discharge. A one-month follow-up visit was prescribed after discharge.

Before the scheduled visit, two weeks after discharge, in March 2024, the patient was admitted directly to our cardiac surgery centre, reporting high fever and acute respiratory failure. A computed tomography (CT) scan of the chest revealed interstitial bilateral pneumonia, and an echocardiogram showed a partial detachment of the mechanical valve along its anterolateral side, necessitating an emergency prosthesis replacement with a biological valve, with a re-sternotomy. Local irrigation with rifamycin solution was also performed. Intraoperative assessment showed numerous endocarditic vegetations with a mammillary lesion along the atrial side and a complete detachment of the anterolateral part of the mitral prosthesis. Blood cultures were still negative, and valve prosthesis samples were then sent to a specialised national microbiology laboratory, as initial cultures remained negative. Four days after emergency prosthetic valve replacement, the patient experienced a spontaneous rupture of the ascending thoracic aorta, necessitating another emergency surgery with a total replacement of the ascending aorta with a collagen-impregnated polyester vascular graft and valve reimplantation. Also, the wall of the aorta was sent to the specialised microbiology laboratory. Broad-range PCR, targeting the 16S rRNA encoding gene, was performed after tissue and prosthetic valve processing. The 16S rRNA gene, widely conserved among bacteria but with variable regions allowing for species-level discrimination, is a standard target for molecular identification in culture-negative infective endocarditis. The amplified DNA was sequenced using Sanger technology, and the resulting sequence was compared against both publicly available databases and internal reference libraries maintained by the institute. Sequence alignment showed >99% identity with Vibrio metschnikovii, confirming its role as the causative pathogen. This molecular diagnosis was crucial, as conventional blood cultures and intraoperative cultures had repeatedly failed to identify the organism. Ciprofloxacin (400 mg intravenously tid) was started. Despite the control of infection markers, neurological damage emerged and worsened, evolving into a vegetative state. Subsequent advanced and focused blood cultures were negative. At one-year post-discharge from the cardiac centre, the patient is hospitalised in a long-term care facility for severe acquired brain injuries.

## 3. Systematic Literature Review

### 3.1. Methodology

**Aim**: The aim of this systematic literature review was to detail further *Vibrio metschnikovii* infections and ascertain the presence of clinical cases of cardiac infection due to *Vibrio metschnikovii*. Thus, the sources of infection, the mechanisms of infection, and the severity of the clinical presentation were investigated.

**Replicability and transparency**: The 27-item Preferred Reporting Items for Systematic reviews and Meta-Analyses (PRISMA) checklist and flowchart [17] were used to conduct the systematic literature review (Appendix A). PROSPERO registration is not applicable, as this was a literature review that used a systematic search.

**Exclusion criteria**: Articles that were not completely retrievable or that were duplicates, articles not written in English, articles not reporting clinical cases, and conference abstracts or reports were excluded.

**Data collection**: On 31 December 2024, V.B. conducted a search on PubMed and EBSCO with the search string “*vibrio*” and “*metschnikovii*”, without time restrictions. Articles were independently pre-screened by A.C. and V.B. for inclusion through title and abstract study. Then, full texts were independently studied by A.C. and V.B. and are summarised in Table 1. Citation cross-checking was performed to identify missing articles. For the included articles, the year of publication, the number of patients involved, country, age and sex, possible source of infection, if mentioned, and clinical presentation were reported.

### 3.2. Results

A PRISMA flow diagram is shown in the Appendix A. Overall, the review yielded 98 articles, reduced to 73 after duplicate removal. Irrelevant articles were removed, with 64 eligible articles remaining, including 1 identified through cross-checking citations. After full-text study, 51 articles dealing only with microbiological, genetical, or environmental issues without reporting on clinical cases were excluded. Finally, 13 articles were included (Table 1). Two articles were included even though they did not report on the age/sex and number of the infected subjects, as they support the role of the pathogen in human gastroenteric infections/contamination [25,28]. The global distribution of the *V. metschnikovii* cases from the literature is reported in Figure 2 with black flags, further indicating the number of cases per country; the site of the discussed case is highlighted with a red flag.

Including the most recent review by Konechnyi Y. et al. [8], 22 cases of clinical manifestations were reported (Table 1). Overall, the patients included 10 males and 5 females, with a median age of 63 years (11–83), plus 1 male neonate aged 5 days. In six cases, patient sex and age were not reported [25]. Overall, there were seven cases of sepsis [8,18,20,26,27,29], of which one was neonatal [20], one with cholecystitis and ascending cholangitis [29], and one with aorto-bifemoral prosthesis graft contamination by aorto-small intestine fistula [8]. In three sepsis cases, echocardiography was performed and came back negative for endocarditis [18,26,27].

Two cases had fatal outcomes [18,27] out of twenty-two total clinical cases; however, if only more severe and systemic cases are included, the mortality rate increases (two fatal outcomes out of seven sepsis cases, of which one was a septic shock [18]).

There were two cases of pneumonia [21,23], of which one was an opportunistic infection linked to intestinal Kaposi’s sarcoma [23]; there were two skin infections [19,22], and eleven cases of diarrhoea [24,25]. A further article reported the presence of *V. metschnikovii* in water contaminated with faecal specimens during a gastroenteritis outbreak in Nigeria [28]. These organ-specific infections are usually clinically mild.

The source of infection could not be documented in any of the cases retrieved but, where ascertained and/or reported, the transmission was due to direct contact with the aquatic environment, the consumption of contaminated food [18,29], or the manipulation of meat products [19,21,22]. Overall, clinical cases did not show recurrent comorbidities and multi-drug resistance.

## 4. Discussion

Based on the literature review, the clinical case described here represents the first documented case of a mitral valve infection due to V. metschnikovii. This report documents the expanding clinical spectrum associated with this rare pathogen and the role it plays in infective endocarditis.

Including the present case of cardiac involvement, a total of 23 clinical cases of V. metschnikovii infection have been described; the male-to-female ratio is 2.2 and the median age is 58 years (min 11–max 83), plus 1 male neonate aged 5 days and 6 cases for which the sex/age of the patients is not indicated. This case, even though it is a singular one, highlights the severity of cardiac involvement, characterised by rapid and extensive tissue destruction and the involvement of the great vessels. Other cases in the literature have shown a severe onset, with signs of sepsis and/or septic shock. They were managed with an immediate initiation of antibiotics, intensive care, aggressive fluid replacement, and vasopressor drugs for hypotension. The antibiotic regimen was adjusted based on the results of the antibiogram, which was not available in our case [18,27].

Enterococci spp., Gram-negative bacilli, HACEK microorganisms (Haemophilus spp, Aggregatibacter actinomycetemcomitans, Cardiobacterium hominis, Eikenella corrodens, and Kingella kingae), and fungi cause most of the remaining cases of endocarditis. Some microorganisms require a longer incubation of blood cultures, others require serodiagnosis (e.g., *Coxiella burnetii*, *Bartonella* spp., *Chlamydia psittaci*, *Brucella* spp.), while others (e.g., *Legionella pneumophila*, *Tropheryma whippelii*) need special culture media or PCR. Persistent negative blood cultures during infective endocarditis may indicate that the infection has been suppressed by previous antibiotic therapy, as it might have been in the case presented. The literature cited does not report particular difficulties in the growth of this pathogen, but its isolation could be difficult, as its growth is inhibited by many of the media used to highlight the most common intestinal pathogens [1,8,13]; the absolute atypicality of localization must certainly be considered, compared to the prevalent gastroenteric involvement of vibrio species.

Another relevant factor that emerged from the overall reconstruction of the case, and with respect to the negativity of the first blood culture, concerns the organisation of microbiology laboratories and remote emergency rooms. A critical limitation in our clinical case is that the initial management of the patient occurred in a primary care facility. Referring first-level laboratories only perform standard microbiological tests and routine blood cultures, with limited expertise in rare pathogens, long-term growth monitoring, and equipment for advanced molecular diagnostics [30,31].

As extensively documented in the literature, culture-negative infective endocarditis represents a diagnostic challenge in clinical practice. Negative blood and tissue cultures can frequently occur. In such scenarios, the continuation of empirical antimicrobial therapy is recommended according to the current guidelines, while the patient should be closely monitored for clinical signs, symptoms, and inflammation markers [32]. In our case, after the initial admission to our centre for valve surgery and the patient’s full recovery, a multidisciplinary inter-hospital consultation was held to discuss the case. The clinicians decided not to pursue further investigation and indicated a follow-up visit and the continuation of antimicrobial treatment at discharge. This approach was adhered to, and only the PCR testing of the mitral tissue prothesis collected during the second surgical intervention allowed for the definitive identification of Vibrio metschnikovii.

The uniqueness of the present case, and the overall rarity of infections caused by this vibrio strain, underscores the importance of considering atypical pathogens in native and prosthetic valve infections and highlights the critical role of molecular diagnostics in guiding effective treatment. This is especially relevant in culture-negative endocarditis. Moreover, in our patient, a clear source and date of the possible infection, together with the route of infection, have not been established. After Vibrio metschnikovii was identified through PCR, the patient’s relatives were once again asked about the patient’s recent travel abroad, gastrointestinal symptoms, or consumption of raw fish or shellfish, but none of these factors were reported, as in the first instance. Occupational exposure was included as a possible source, but there were no reported skin injuries or other traumas that could have served as an entry site for the pathogen. Although cases involving other Vibrio species have been documented with possible occupational links due to pond-cultivated fish exposure, the infections are usually limited to the skin and/or soft tissue, and bacteraemia spread is due to individual predisposing conditions [6,33]. None of those conditions were documented. Moreover, no similar cases, nor a significant presence of the pathogen in the Mediterranean Sea, have been reported [34,35,36].

The diagnostic challenges in this case were significant because standard techniques of infectious disease investigation failed and advanced molecular diagnostics were necessary. Although the identification of Vibrio metschnikovii by PCR testing was crucial in guiding antimicrobial therapy, unfortunately, the technology of biomolecular testing did not allow for an antibiogram, and the choice of antibiotic therapy was based on the literature, with ciprofloxacin. The treatment regimen for the patient initially included broad-spectrum antibiotics, which were essential in managing his rapidly deteriorating condition. It should be noted that 22% of the Vibrio isolates in Falco et al. were resistant to two of the families of antibiotics most used in clinical practice (third-generation cephalosporins and aminoglycosides). Those data are also supported by data from aquatic isolates and from samples from infections in both animals and humans, even though no multidrug resistance was observed [2]. The case recently reported by Konechnyi et al. showed V. metschnikovii’s resistance to ceftriaxone, while in another clinical case, poor sensibility to third-generation cephalosporins was documented [8,19]. Therefore, switching to ciprofloxacin might have been pivotal in the recovery of the infection.

Additionally, the spontaneous rupture of the ascending thoracic aorta in the patient further complicates the clinical history and recovery, exacerbating the severity of this infection.

Given the increasing incidence of non-cholera Vibrio infections worldwide, more research is needed to understand the environmental and genetic factors contributing to their pathogenicity [37]. Studies focusing on the mechanisms of non-cholera Vibrio adhesion to prosthetic devices, the role of biofilms in persistent infections, and the development of rapid diagnostic tools are essential for improving patient outcomes [6,38,39].

Collaborative efforts between environmental scientists, epidemiologists, and healthcare providers can help develop comprehensive approaches to manage and prevent non-cholera Vibrio infections.

## 5. Limitations

The general limitations of systematic reviews and case reports are biases of publication due to negative outcomes or lack of microbiological ascertainment, depending also on the availability of advanced testing methods and on laboratory expertise [40]. The unavailability of an antibiogram in the specific case described limits the considerations of the efficacy of the selected antibiotic regimen.

## 6. Recommendations for Culture-Negative Endocarditis

Investigate further patient history (illicit drugs, medication, probiotics, travel, life and recreation habits).Perform a clinical examination to identify potential risk factors for specific microorganisms, serologies for specific microorganisms, or non-infectious miming causes.Verify proper management and performance of blood samples and tissue samples for culture after surgery.Prolonged incubation for up to 14 days may be beneficial for detecting certain organisms.Refer to specialised microbiology laboratories with advanced diagnostic methods, like MALDI-TOF, broad-range PCR, or targeted metagenomic sequencing.

## 7. Conclusions

To the best of our knowledge, this case represents the first documented case of a mitral valve infection and cardiac tissue involvement caused by Vibrio metschnikovii.

This case highlights the critical issue of the identification of rare pathogens in the clinical setting. It adds an incipient insight into the clinical management of Vibrio metschnikovii infections and expands the known clinical spectrum of this rare pathogen. This case report brings into view the diagnostic complexity of endocarditis in a globalised context, characterised by atypical pathogens with specific growth conditions and various forms of resistance. This becomes even more difficult once broad-spectrum antibiotic therapy has been initiated.

Furthermore, this case underscores the importance of investigating the possible routes and sources of infection and to define the prevalence of these pathogens, driven by global warming.

## Figures and Tables

**Figure 1 clinpract-15-00118-f001:**
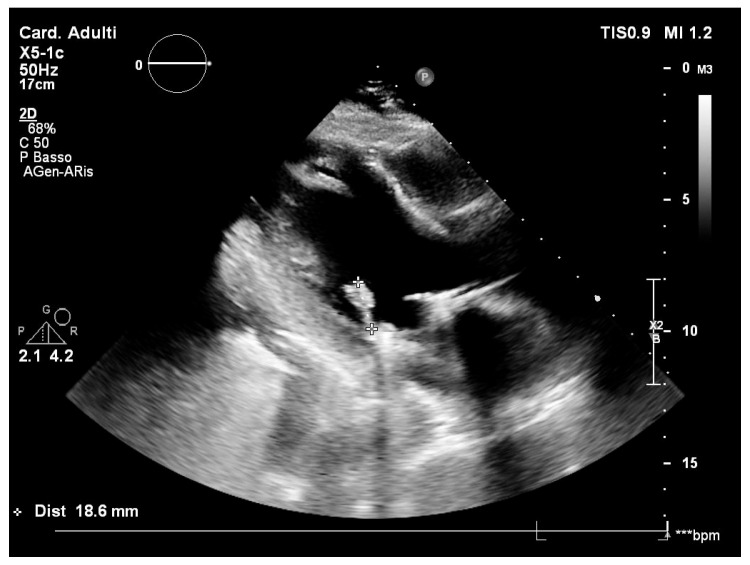
Transoesophageal echocardiography with mitral valve vegetations. *** transoesophageal.

**Figure 2 clinpract-15-00118-f002:**
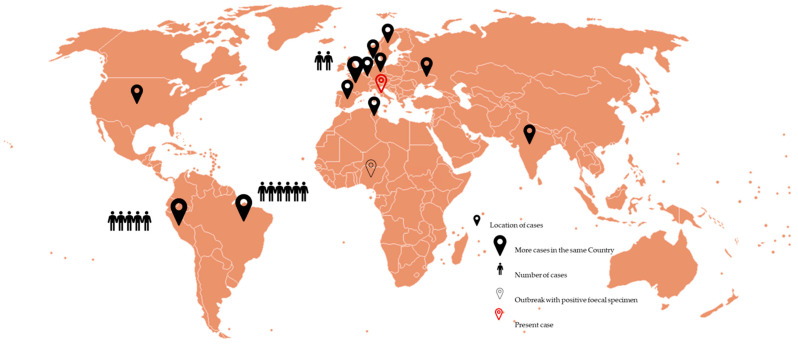
Global distribution of cases after systematic literature review.

**Table 1 clinpract-15-00118-t001:** Literature review on infection due to *Vibrio metschnikovii* and clinical cases described.

Author and Year [Reference]	Number of Patients	Country	Age (Years)	Sex (M Male, F Female)	Source of Infection (If Available)	Clinical Diagnosis	Other Information
Konechnyi, 2021 [8]	1	Ukraine	70	M	Patient denied having contact with the sea or consuming marine products. Source not identified	Sepsis with aorto-bifemoral graft prosthesis infection due to aorto-small intestine fistula	Aorto-bifemoral graft pseudoaneurysm and aorto-small intestine fistula could be the cause for bacteriaemia
Jensen, 2014 [18]	1	Denmark	78	M	Probably food ingestion	Gastroenteritis and septic shock. Death	Negative echocardiography
Pariente Martìn, 2008 [19]	1	Spain	49	F	Moved five months prior from Uruguay, with a 7-year history of fibromyalgia and infected leg ulcers	Wound infection and ulcers in both legs, with chronic lymphoedema	_
Prasad, 2005 [20]	1	India	5 days	M	Possible perinatal transmission but not supported	Neonatal sepsis	_
Wallet, 2005 [21]	1	France	63	M	Retired carpenter with no contact with domestic or wild animals and negative recent history of diarrhoea. Source not identified	Pneumonia	_
Linde, 2004 [22]	1	Germany	64	M	Probably a zoonotic source; patient worked as a butler	Wound infection	_
Ben Rejeb, 2001 [23]	1	Tunisia	52	M	Source not identified	Opportunistic pneumonia in primary digestive tract Kaposi sarcoma	_
Dalsgaard, 1996 [24]	5	Peru	15, 12, 12, 20, and 11	4 M, 1 F	Source not identified	Acute diarrhoea, with two cases showing moderate dehydration	Outbreak of diarrhoea associated with V. metschnikovii
Magalhães, 1996 [25]	6	Brazil	na	na	Three patients denied exposure to seafood. Source not identified	Diarrhoea and positivity in faecal specimens	Assessment of 4000 diarrheal faecal specimens between 1992 and 1993 with 73 Vibrio isolates and 6 of V. metschnikovii
Hardardottir, 1994 [26]	1	Sweden	83	F	Patient was not recently abroad and did not consume raw seafood. Source not identified	Sepsis with concomitant Staphylococcus hominis and Escherichia coli infections	Negative echocardiography
Hansen, 1993 [27]	2	Belgium and France	80 and 70	F, M	No travel or seafood consumption. Source not identified	Sepsis, one of which with wound infection. One patient dead	Negative echocardiography
Bitto, 1992 [28]	na	Nigeria	na	na	Probable water contamination after a festival with infected visitors	Outbreaks of gastroenteritis with the presence of V. metschnikovii in water and faecal specimens	_
Jean-Jacques, 1981 [29]	1	US	82	F	Probably long-term gallbladder carriage after previous sea contact or eating seafood	Sepsis with cholecystitis and ascending cholangitis	_

Legend: na means not available.

## Data Availability

Data is available from the corresponding author on request.

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
