# Peer review of "First Case of Infective Endocarditis Caused by Vibrio metschnikovii: Clinico-Diagnostic Complexities and a Systematic Literature Review"

_clinpract, 2025, doi:10.3390/clinpract15070118_

Round 1
Reviewer 1 Report
Comments and Suggestions for Authors
This manuscript of Carrozzo et al. is an interesting attempt to present a clinical case of infective endocarditis by Vibrio metschnikovii along with a systematic review of the existing literature.
The case report presented is rare and of high interest and the use of PRISMA checklist provides an important guidance but the structure and the general presentation should be improved for the comprehension and readability to be enhanced.
The English language and the formality should also be improved.
Abstract: The main idea of the manuscript and the basic information are included but I recommend it to be restructured along with the addition of more information in the ‘background’ subsection and the limitation of ‘case report’ description and ‘discussion’ subsection to the most important and absolutely necessary information.
- Line 21-23: Please rephrase and provide a better and more comprehensible description of these cases.
- Line 28-29: The meaning of this sentence is not absolutely clear, I recommend to formulate it differently.
- I recommend the ‘Discussion’ subsection to be restructured and rephrased including more important information of the discussion along with a more accurate presentation. For example, I suggest not to present in that extend the patient’s medical history while ignoring more important information regarding the medical procedures and the outcome of this clinical case.
- Line 37-38: I suggest this first sentence to be placed in the end of the ‘conclusion’ subsection.
Introduction: Please add more medical information including possible complications of endocarditis and suggested therapy.
Case report:
- Line 67: Please rephrase ‘We report the case of a 28-year-old…’
- Line 69-70: The meaning of this sentence is not absolutely clear. Please present it better – who used the protective equipment and for what purpose?
- Line 71: Rephrase ‘denied taking illegal drugs’
- Line 72: Please replace ‘went’ with ‘admitted’
- Line 92-94: I recommend to rephrase the whole sentence. For example: ‘After 10 days of hospitalization in the ICU, the patient’s clinical condition was improved and he was finally discharged’
- Line 95: Please replace ‘per iv’ with ‘intravenously’
- Line 95-96: I suggest to rephrase the whole sentence for better comprehension.
- Line 97-98: Please re-arrange this sentence and also replace ‘presented’ with ‘was admitted’ and ‘with’ with ‘reporting’ for a more formal and accurate presentation.
Systemic literature review – Methodology: The use of PRISMA guidelines is a positive choice. For the ‘methodology’ section I recommend the creation of subsection such as ‘inclusion criteria’, ‘exclusion criteria’, ‘data collection’, etc for a more enhanced structure and better comprehension.
Systemic literature review – Results:
- Figure 1: It would be useful for the reader to explain shortly this figure.
- Line 150-159: Please rewrite the whole paragraph to be more comprehensible and readable. Provide more details.
- Table 1: I recommend not to include the present case as this is a table reviewing existing literature. Also, I the last line of the table, the number ‘58’ is referring to the median age of patients mentioned? If so, please add the range.
Discussion
- Line 173: Please replace ‘resulted’ with ‘were conducted’.
- Line 175: Please add the range of the median age.
- Line 177-178: The aim could also be mentioned in the abstract and the introduction of this manuscript.
- Line 179-180: Please remove limitations to the end of this section.
- Line 180-185: I recommend to rewrite this paragraph in greater extent, adding the references of every case that is mentioned and comparing the findings with your case report. It should also be sited below in discussion.
- Line 206-212: This paragraph is also referred to limitations, I suggest it to be sited below in discussion.
- Line 213: Please erase the words ‘those limitations are particularly crucial because’.
- Line 231: In this sentence you mean that the medical and travel history was re-examined? Please determine.
- Line 258-259: Is there any reference for this statement?
Conclusion:
- Line 274-275: ‘One lesson that can be reiterated…’ please rephrase with more formality.
The expressions and the words used should be more formal and better presented so that the comprehension and readability to be enhanced.
Author Response
We would like to thank the Editors and Reviewers for the comments and suggestions which gave us the opportunity to make the manuscript more consistent and clearer.
We have answered the reviewers in the specific online section by rewriting Reviewers’ comments in bold, followed by authors answers. In the manuscript, changes are tracked with Word track processor (the text added resulted in different colours, while the text removed is coloured and crossed out). We report here all editors/reviewers' comments and authors’ answers.
This manuscript of Carrozzo et al. is an interesting attempt to present a clinical case of infective endocarditis by Vibrio metschnikovii along with a systematic review of the existing literature.
The case report presented is rare and of high interest and the use of PRISMA checklist provides an important guidance but the structure and the general presentation should be improved for the comprehension and readability to be enhanced.
The English language and the formality should also be improved.
Many thanks, we have extensively revised the text.
Abstract: The main idea of the manuscript and the basic information are included but I recommend it to be restructured along with the addition of more information in the ‘background’ subsection and the limitation of ‘case report’ description and ‘discussion’ subsection to the most important and absolutely necessary information.
Line 21-23: Please rephrase and provide a better and more comprehensible description of these cases.
Line 28-29: The meaning of this sentence is not absolutely clear, I recommend to formulate it differently.
I recommend the ‘Discussion’ subsection to be restructured and rephrased including more important information of the discussion along with a more accurate presentation. For example, I suggest not to present in that extend the patient’s medical history while ignoring more important information regarding the medical procedures and the outcome of this clinical case.
Line 37-38: I suggest this first sentence to be placed in the end of the ‘conclusion’ subsection.
We have revised the text, integrating your suggestions and other reviewers’ ones, also considering words limitation.
Introduction: Please add more medical information including possible complications of endocarditis and suggested therapy.
Many thanks, we have revised the text and sections’ content, integrating your suggestions and other reviewers’ ones.
Case report:
Line 67: Please rephrase ‘We report the case of a 28-year-old…’
Line 69-70: The meaning of this sentence is not absolutely clear. Please present it better – who used the protective equipment and for what purpose?
Line 71: Rephrase ‘denied taking illegal drugs’
Line 72: Please replace ‘went’ with ‘admitted’
Line 92-94: I recommend to rephrase the whole sentence. For example: ‘After 10 days of hospitalization in the ICU, the patient’s clinical condition was improved and he was finally discharged’
Line 95: Please replace ‘per iv’ with ‘intravenously’
Line 95-96: I suggest to rephrase the whole sentence for better comprehension.
Line 97-98: Please re-arrange this sentence and also replace ‘presented’ with ‘was admitted’ and ‘with’ with ‘reporting’ for a more formal and accurate presentation.
We have revised the text answering all those indications for improvement.
Systemic literature review – Methodology: The use of PRISMA guidelines is a positive choice. For the ‘methodology’ section I recommend the creation of subsection such as ‘inclusion criteria’, ‘exclusion criteria’, ‘data collection’, etc for a more enhanced structure and better comprehension.
We have revised the section and highlighted in bold the sub-sections.
Systemic literature review – Results:
Figure 1: It would be useful for the reader to explain shortly this figure.
We have added an explaining sentence.
Line 150-159: Please rewrite the whole paragraph to be more comprehensible and readable. Provide more details.
We have revised the section.
Table 1: I recommend not to include the present case as this is a table reviewing existing literature. Also, I the last line of the table, the number ‘58’ is referring to the median age of patients mentioned? If so, please add the range.
We have revised the text and removed our case. The synthesis of results from literature review and our case was discussed in the discussion section.
Discussion
Line 173: Please replace ‘resulted’ with ‘were conducted’.
Line 175: Please add the range of the median age.
And
Line 213: Please erase the words ‘those limitations are particularly crucial because’.
Line 231: In this sentence you mean that the medical and travel history was re-examined? Please determine.
We have revised the text.
Line 177-178: The aim could also be mentioned in the abstract and the introduction of this manuscript.
We have added in the abstract and in the introduction.
Line 179-180: Please remove limitations to the end of this section.
And
Line 206-212: This paragraph is also referred to limitations, I suggest it to be sited below in discussion.
We have revised the text and more conveniently moved the limitation after discussion in a comprehensive paragraph.
Line 180-185: I recommend to rewrite this paragraph in greater extent, adding the references of every case that is mentioned and comparing the findings with your case report. It should also be sited below in discussion.
We have revised the text accordingly.
Line 258-259: Is there any reference for this statement?
We simply mention the severity of the clinical case with the cardiac involvement. We have removed the sentence.
Conclusion:
Line 274-275: ‘One lesson that can be reiterated…’ please rephrase with more formality.
Thanks, we have revised the text accordingly.
Comments on the Quality of English Language
The expressions and the words used should be more formal and better presented so that the comprehension and readability to be enhanced. 
Many thanks, we have extensively revised the text.
Sincerely
vb
Reviewer 2 Report
Comments and Suggestions for Authors
The manuscript by Carrozzo et al. describes a case report of a 28-year-old man with endocarditis caused by Vibrio metschnikovii, representing the underlying cause of his condition. This manuscript is very interesting, as it highlights the importance of screening for rare pathogens in patients presenting with endocarditis and negative blood or tissue cultures. The manuscript is well written, and the literature review and data are well presented. I would only suggest expanding the Results section by providing a brief description of the studies summarized in Table 1.
Author Response
REV. 2
We would like to thank the Editors and Reviewers for the comments and suggestions which gave us the opportunity to make the manuscript more consistent and clearer.
We have answered the reviewers in the specific online section by rewriting Reviewers’ comments in bold, followed by authors answers. In the manuscript, changes are tracked with Word track processor (the text added resulted in different colours, while the text removed is coloured and crossed out). We report here all editors/reviewers' comments and authors’ answers.
The manuscript by Carrozzo et al. describes a case report of a 28-year-old man with endocarditis caused by Vibrio metschnikovii, representing the underlying cause of his condition. This manuscript is very interesting, as it highlights the importance of screening for rare pathogens in patients presenting with endocarditis and negative blood or tissue cultures. The manuscript is well written, and the literature review and data are well presented. I would only suggest expanding the Results section by providing a brief description of the studies summarized in Table 1.
Many thanks, we have expanded the results section regarding the literature review data.
sincerely
vb
Reviewer 3 Report
Comments and Suggestions for Authors
The manuscript is from a scientific viewpoint excellent.
I have one remark. The authors need to briefly describe the methods they used to examine the staff they discovered the Vibrio.
Author Response
We would like to thank the Editors and Reviewers for the comments and suggestions which gave us the opportunity to make the manuscript more consistent and clearer.
We have answered the reviewers in the specific online section by rewriting Reviewers’ comments in bold, followed by authors answers. In the manuscript, changes are tracked with Word track processor (the text added resulted in different colours, while the text removed is coloured and crossed out). We report here all editors/reviewers' comments and authors’ answers.
The manuscript is from a scientific viewpoint excellent.
I have one remark. The authors need to briefly describe the methods they used to examine the staff they discovered the Vibrio.
Many thanks, we have added a sentence on the microbiological methodology in the case report section.
sincerely
vb
Reviewer 4 Report
Comments and Suggestions for Authors
Thank you for the opportunity to review your manuscript. I appreciate the effort and thought that went into your research and the clarity with which you have presented your findings. Your work addresses an important topic in the field, yet I have two issues to be addressed
First: The reader may infer that this review focuses on cardiac complications associated with Vibrio metschnikovii; however, Table 1 indicates that only one case—your recently reported case—involved cardiac manifestations, please add more clarification.
A second suggestion would be to include clinical implications or recommendations for practice. For instance, in cases where endocarditis is suspected, it would be helpful to discuss the recommended duration of blood culture incubation to ensure detection of rare pathogens such as Vibrio metschnikovii.
best regards
Author Response
We would like to thank the Editors and Reviewers for the comments and suggestions which gave us the opportunity to make the manuscript more consistent and clearer.
We have answered the reviewers in the specific online section by rewriting Reviewers’ comments in bold, followed by authors answers. In the manuscript, changes are tracked with Word track processor (the text added resulted in different colours, while the text removed is coloured and crossed out). We report here all editors/reviewers' comments and authors’ answers.
Thank you for the opportunity to review your manuscript. I appreciate the effort and thought that went into your research and the clarity with which you have presented your findings. Your work addresses an important topic in the field, yet I have two issues to be addressed
First:  The reader may infer that this review focuses on cardiac complications associated with Vibrio metschnikovii; however, Table 1 indicates that only one case—your recently reported case—involved cardiac manifestations, please add more clarification.
Many thanks. We have revised the introduction and better explained our aim.
A second suggestion would be to include clinical implications or recommendations for practice. For instance, in cases where endocarditis is suspected, it would be helpful to discuss the recommended duration of blood culture incubation to ensure detection of rare pathogens such as Vibrio metschnikovii. best regards
Many thanks. We have added a brief paragraph with practical point recommendations.
sincerely
vb
Round 2
Reviewer 1 Report
Comments and Suggestions for Authors
After the modifications suggested, the manuscript has been improved considering its presentation, structure and efficacy.
Author Response
After the modifications suggested, the manuscript has been improved considering its presentation, structure and efficacy.
Many thanks, sincerely, vittorio bolcato